# Scalable and switchable CO₂-responsive membranes with high wettability for separation of various oil/water systems

Yangyang Wang[1,8], Shaokang Yang[2,8], Jingwei Zhang[3,8], Zhuo Chen[3], Bo Zhu[4], Jian Li[5], Shijing Liang[6], Yunxiang Bai[1], Jianhong Xu[3], Dewei Rao[2], Liangliang Dong[1]✉, Chunfang Zhang[1] & Xiaowei Yang[7]

Smart membranes with responsive wettability show promise for controllably separating oil/water mixtures, including immiscible oil-water mixtures and surfactant-stabilized oil/water emulsions. However, the membranes are challenged by unsatisfactory external stimuli, inadequate wettability responsiveness, difficulty in scalability and poor self-cleaning performance. Here, we develop a capillary force-driven confinement self-assembling strategy to construct a scalable and stable CO₂-responsive membrane for the smart separation of various oil/water systems. In this process, the CO₂-responsive copolymer can homogeneously adhere to the membrane surface by manipulating the capillary force, generating a membrane with a large area up to 3600 cm² and excellent switching wettability between high hydrophobicity/ underwater superoleophilicity and superhydrophilicity/underwater superoleophobicity under CO₂/N₂ stimulation. The membrane can be applied to various oil/water systems, including immiscible mixtures, surfactant-stabilized emulsions, multiphase emulsions and pollutant-containing emulsions, demonstrating high separation efficiency (>99.9%), recyclability, and self-cleaning performance. Due to robust separation properties coupled with the excellent scalability, the membrane shows great implications for smart liquid separation.

The ability to manipulate the surface properties, e.g., wettability and liquid repellency, is of great significance in both fundamental research and application[1–5]. The classical example of this is membrane-based oil-water separation, in which a membrane with special wettability allows one phase (for example, oil) to penetrate while blocking the other phase (for example, water)[6–8]. Nevertheless, the single and unalterable wettability of most traditional superwetting membranes has severely limited their applications to one certain case (either water removing or oil removing). Considering the complexity of actual oil/water mixtures with the coexistence of different types of immiscible and emulsified

[1]Key Laboratory of Synthetic and Biological Colloids, Ministry of Education, School of Chemical and Material Engineering, Jiangnan University, 214122 Wuxi, P. R. China. [2]School of Materials Science and Engineering, Jiangsu University, 212013 Zhenjiang, P. R. China. [3]The State Key Laboratory of Chemical Engineering, Department of Chemical Engineering, Tsinghua University, 100084 Beijing, P. R. China. [4]Key Laboratory of Eco-textiles, Ministry of Education, Jiangnan University, 214122 Wuxi, P. R. China. [5]Laboratory of Environmental Biotechnology, Jiangsu Engineering Laboratory for Biomass Energy and Carbon Reduction Technology, Jiangsu Key Laboratory of Anaerobic Biotechnology, School of Environmental and Civil Engineering, Jiangnan University, 214122 Wuxi, P. R. China. [6]National Engineering Research Center of Chemical Fertilizer Catalyst, Fuzhou University, Fuzhou 350002, P. R. China. [7]School of Chemistry and Chemical Engineering, Shanghai Jiao Tong University, Shanghai 200240, P. R. China. [8]These authors contributed equally: Yangyang Wang, Shaokang Yang, Jingwei Zhang. ✉e-mail: liangliangdong@jiangnan.edu.cn

mixtures, the development of smart membranes with tunable wettability for application in multitype oil/water separation is highly desired.

Inspired by cell membranes with stimuli-responsive channels for self-regulating their mass transfer and interfacial properties in response to changes in environmental conditions[9,10], artificial wettability-switchable membranes have become a new frontier in the field of intelligent oil-water separation. Compared with traditional water removing or oil removing membranes, these artificial membranes can undergo a structural, morphological, or molecular conformational switch to tune the surface wettability and liquid transport channels in response to external stimuli (e.g., temperature, pH, electricity, light, magnetic field or ions); as a result, controllable oil/water separation is achieved. To date, much work on artificial wettability-switchable membranes has been reported for the controllable separation of oil/water mixtures[11–21]. However, some persistent challenges remain unresolved. On the one hand, most existing wettability-switchable membranes suffer from complex and expensive fabrication processes (e.g., chemical grafting and layer-by-layer self-assembly)[22]. These disadvantages not only hinder their compatibility with large-scale applications but also lead to a sparse or inhomogeneous presence of responsive moieties in the membrane, which is accompanied by inadequate responsiveness of surface wettability and deficient separating controllability of various emulsions. On the other hand, there are major limitations in applying the aforementioned triggers, including economic and environmental costs and product contamination. For instance, the stimuli of high-energy electricity and heat may damage the system to a certain extent[23]. The stimuli of light, magnetic or mechanical fields suffer from the limitation of penetration depth. When using pH, ionic, enzyme, or redox as triggers, the inevitable generation of byproducts accompanied by the repeated addition of chemical agents generates a complex and weak cycling process with low sensitivity depletion[10].

In recent years, there has been growing interest in $CO_2$-responsive materials and systems. Compared with other stimuli, $CO_2$ is nontoxic, inexpensive, does not accumulate chemical species and is easily added to or removed from systems under the operation condition[24–32]; thus, $CO_2$ is a promising candidate for the design of wettability-switchable membranes. For instance, Yuan et al. fabricated a $CO_2$-responsive nanofibrous membrane for highly efficient oil/water separation[33]. Under alternating $CO_2/N_2$ stimulation, this nanofibrous membrane displayed reversible regulation of surface oil/water wettability, thus achieving highly controllable separation of immiscible oil/water mixtures. A porous membrane with an open-cell structure and $CO_2$ switchable wettability was reported by Zhu et al. through continuous water-in-oil high internal phase emulsion templates[34]. The wettability of this membrane could be switched between hydrophobic or superoleophilic and hydrophilic or underwater superoleophobic through drying or $CO_2$ treatment, resulting in a high separation efficiency >96 wt% for both chloroform/water and water/hexane systems. Despite these great advances, current $CO_2$-responsive membranes remain effective only in separating immiscible oil-water mixtures, and the relevant research on $CO_2$-responsive membranes for various stable emulsion systems remains unexplored. In addition, since the fabrication strategies of $CO_2$-responsive membranes are similar to those of other stimuli-responsive membranes, some critical issues, including complex fabrication, low productivity, and difficult scale-up, remain inescapable, which means that the current fabrication strategies are only limited to the laboratory and are difficult to industrialize. Therefore, a facile and low-cost route to fabricate $CO_2$-responsive membranes remains an elusive challenge for exploiting their application potential in complicated emulsion systems, and research efforts in this direction are critically needed.

Here, inspired by the capillary phenomenon in nature, we present a conceptual design strategy for the fabrication of $CO_2$-responsive

membranes through a capillary force-driven confinement self-assembling (CFCS) method. This method is achieved by parallelly stacking two hydrophobic substrates with clearance to form the capillary force to drive the $CO_2$-responsive polymer solution (poly(diethylaminoethyl methacrylate-co-methyl methacrylate, PMMA-co-PDEAEMA) into the confined area, followed by self-assembly in situ on the surface and inside of fabric. By manipulating the capillary force, PMMA-co-PDEAEMA can homogeneously adhere to the fabric to enhance the switching ability of the membrane surface wettability. In this work, the copolymer assembly behaviors and surface wettability of the membrane during the CFCS process were first studied. Then, the gas switchable surface wettability of the resultant membranes was thoroughly investigated by alternating the $CO_2/N_2$ stimulation levels and the ratio of MMA/DEAEMA in the copolymer. Finally, the gas-tunable separation performance of the membranes for various oil/water systems, especially surfactant-stabilized O/W and W/O emulsions and multiphase emulsion mixtures, as well as the gas-tunable self-cleaning performance for various contaminants, were systematically investigated. Considering both fundamental research and industrial applications, we anticipate that the proposed CFCS method will provide an effective strategy for realizing industrial-scale production of stimuli-responsive membranes.

## Results and discussion
### Fabrication and structural characterization
The fabrication process of the $CO_2$-responsive membrane through the CFCS method is schematically depicted in Fig. 1A. A series of PMMA-co-PDEAEMA copolymers synthesized by radical copolymerization reactions (Supplementary Fig. 1 and Supplementary Table 1) were chosen as $CO_2$-responsive polymers in this work, in which the PDEAEMA segments endow the copolymers with high sensitivity to $CO_2$, while the PMMA segments that are insoluble in water, prevent falling off of copolymers from fabric substrate under $CO_2$ stimulation. Next, a piece of polyester fabric was fixed inside of the gap formed by two pieces of superimposed acrylic plates (Fig. 1B). After that, 10 wt% of the copolymer solution (THF as solvent) was slowly injected into the edge of the gap. Then, under capillary force, the solution was spread in confined space formed by gap width, and self-assembly occurred in situ on the surface and inside of the fabric. After thermal treatment, a PMMA-co-PDEAEMA-coated fabric membrane, namely, PPFM, was successfully constructed. In the CFCS process, the gap width and wettability of the plate are two critical factors determining the morphology and wetting behavior of the as-prepared PPFM, as the former determines the capillary force and the latter determines the direction of the capillary force according to the Young's and Laplace equations[35,36]. Therefore, in this work, the wettability of the plate was investigated by measuring its contact angle (CA) under the copolymer solution. Figure 1B and Supplementary Fig. 2 depict the dynamic CA transformation process over time. Notably, the copolymer droplet collapses and spreads out completely in a short time, demonstrating the excellent wettability of the copolymer solution on the surface of the acrylic plate. In this way, the head of the copolymer solution inside the gap possesses a concave meniscus (Supplementary Fig. 3), keeping the direction of capillary force toward the interior of the gap, which is beneficial for transporting the solution in the gap. The effect of the gap width on the capillary force is shown in Supplementary Fig. 4. On the one hand, according to the Laplace equation, capillary force is inversely proportional to capillary radius (the gap width in this case)[33]. Therefore, increasing the gap width could lead to a continuous decline in the capillary force in theory, which in turn results in insufficient force to drive diffusion of the solution in the capillary. On the other hand, with increasing gap width, gravitational effects are not negligible and have an important effect on the diffusion behavior of the solution above and below the fabric substrate. That is, the solution above the fabric substrate (referred to as the A side) tends to be deposited on the fabric, while the

solution below (referred to as the B side) tends to be away from the fabric under gravity, causing heterogeneous copolymer assembly behaviors and surface wettability on the two sides. To further confirm this hypothesis, scanning electron microscopy (SEM) was conducted to characterize the morphologic evolution of the two sides. As shown in Fig. 1C and Supplementary Fig. 5, when increasing the gap width from 150 μm to 300 μm, the fiber skeletons and 3D braided structures of the A side are gradually covered by the copolymer, especially at a gap width of 300 μm, as they are completely covered. In contrast, this structural information can always be observed on the B side over the entire range of gap widths. Cross-sectional SEM images (Supplementary Fig. 6) and EDS line-scan (Supplementary Figs. 7 and 8) further confirm this morphology difference, as the thickness of the copolymer layer on the A side shows a significant increase, while that on the B side only shows a slight increase with increasing gap width (Supplementary Fig. 7). In addition, water contact angle (WCA) analysis was conducted to further investigate the influence of gap width on the surface wetting behaviors of the two sides. As shown in Supplementary Fig. 9, the difference between the WCAs of the two sides jumps from ~0° to ~24° with increasing gap width, confirming that a large gap width is unfavorable for the formation of uniform surface wettability on the two sides. From Fig. 1C and Supplementary Fig. 5, it can also be found that increasing the gap width decreases the WCAs of both sides. For instance, the WCA of the B side is 146.2° at a gap width of 150 μm, while

the value dramatically decreases to 97.8° at a gap width of 300 μm. This phenomenon mainly occurs because the thick copolymer layer at a large gap width smooths the 3D surface topology of the as-prepared PPFM, thus resulting in a decline in surface roughness. COMSOL simulations were performed to further investigate the assembly behaviors of the copolymers on each side of the fabric substrate, and this information was used to elucidate the morphology difference of the PPFM (details of the simulations are available in the SI). Figure 1D and Supplementary Figs. 10 and 11 show the velocity and pressure fields of the copolymer solution on each side under different gap widths during the CFCS process. Increasing the gap width causes a distinctly asymmetric distribution of both the velocity and pressure on the two sides, which agrees well with the experimental results.

According to the aforementioned results, the optimal gap width was set to 150 μm to reach the ideal morphology and stable surface wettability. The detailed chemical compositions of the as-prepared PPFM at a gap width of 150 μm were characterized by energy-dispersive X-ray spectroscopy (EDX) element mapping (Supplementary Fig. 12), Fourier transform infrared (FT-IR) spectrometry (Supplementary Fig. 13), and X-ray photoelectron spectroscopy (XPS) (Supplementary Fig. 14). As shown in Supplementary Fig. 9, compared with the pristine fabric, a new element, N, derived from the PMMA-*co*-PDEAEMA copolymer can be observed and uniformly distributed in the PPFM, indicating successful fabrication of PPFM. The magnified

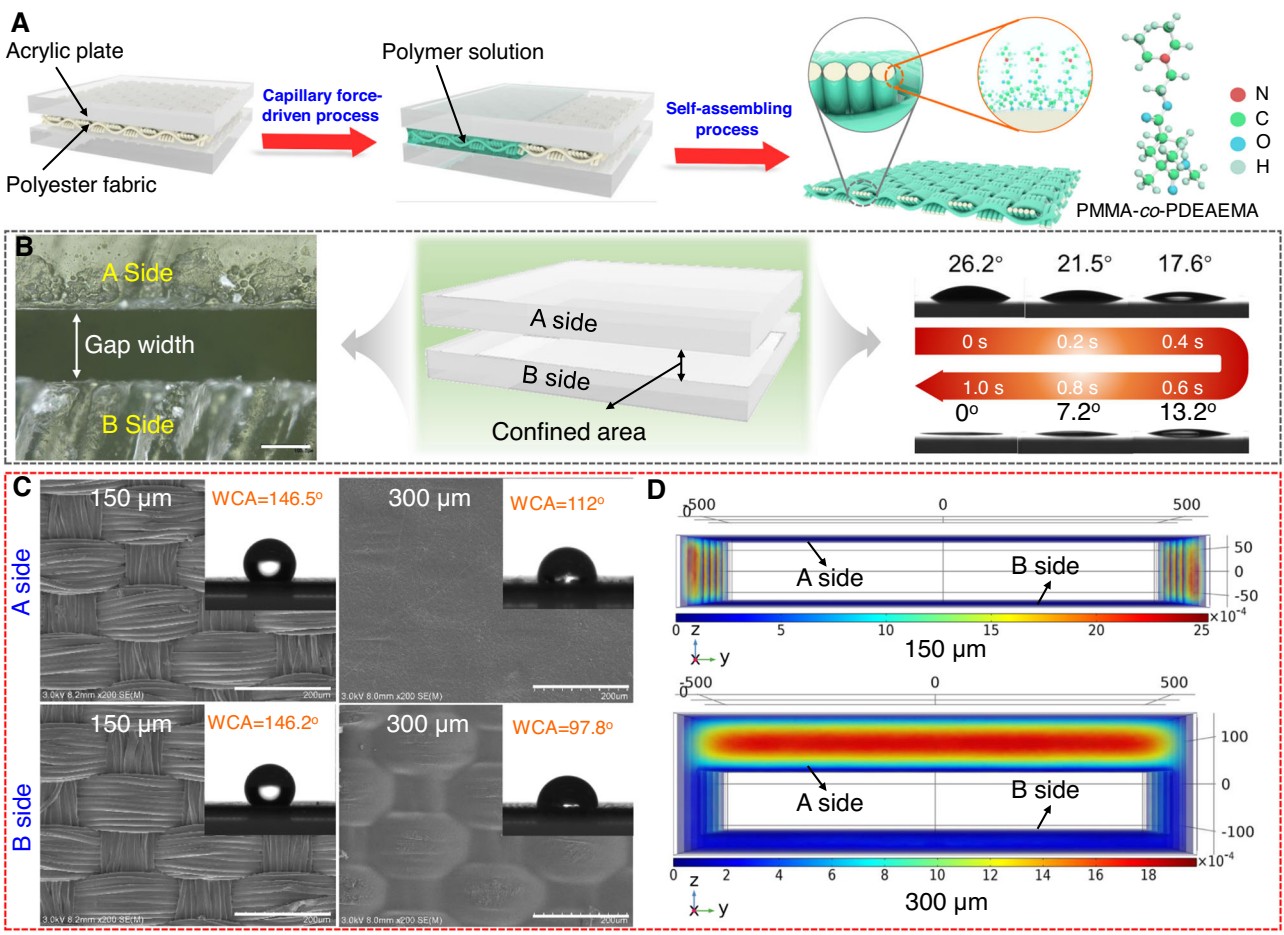

**Fig. 1 | Design of PPFM. A** Schematic illustration of the fabrication process of PPFM through the capillary force-driven confinement self-assembling method. **B** Schematic illustration (middle) and optical image (left) of the formation of a confined area caused by two pieces of superimposed acrylic plates. Dynamic CA transformation (right) of PMMA-*co*-PDEAEMA copolymer droplet on the surface of the acrylic plate over time. The used droplet is a 10 wt% PMMA-*co*-PDEAEMA

copolymer (MMA/DEAEMA ratio of 0.5) solution. **C** SEM images and WCA of the as-prepared PPFM surfaces on two sides. The scale bar is 200 μm. **D** Velocity field of the copolymer solution on each side under different gap widths based on the COMSOL simulation. With increasing gap width, the velocity of the copolymer solution on the A side is larger than that on the B side, resulting in a larger accumulation of the PMMA-*co*-PDEAEMA copolymer on the A side.

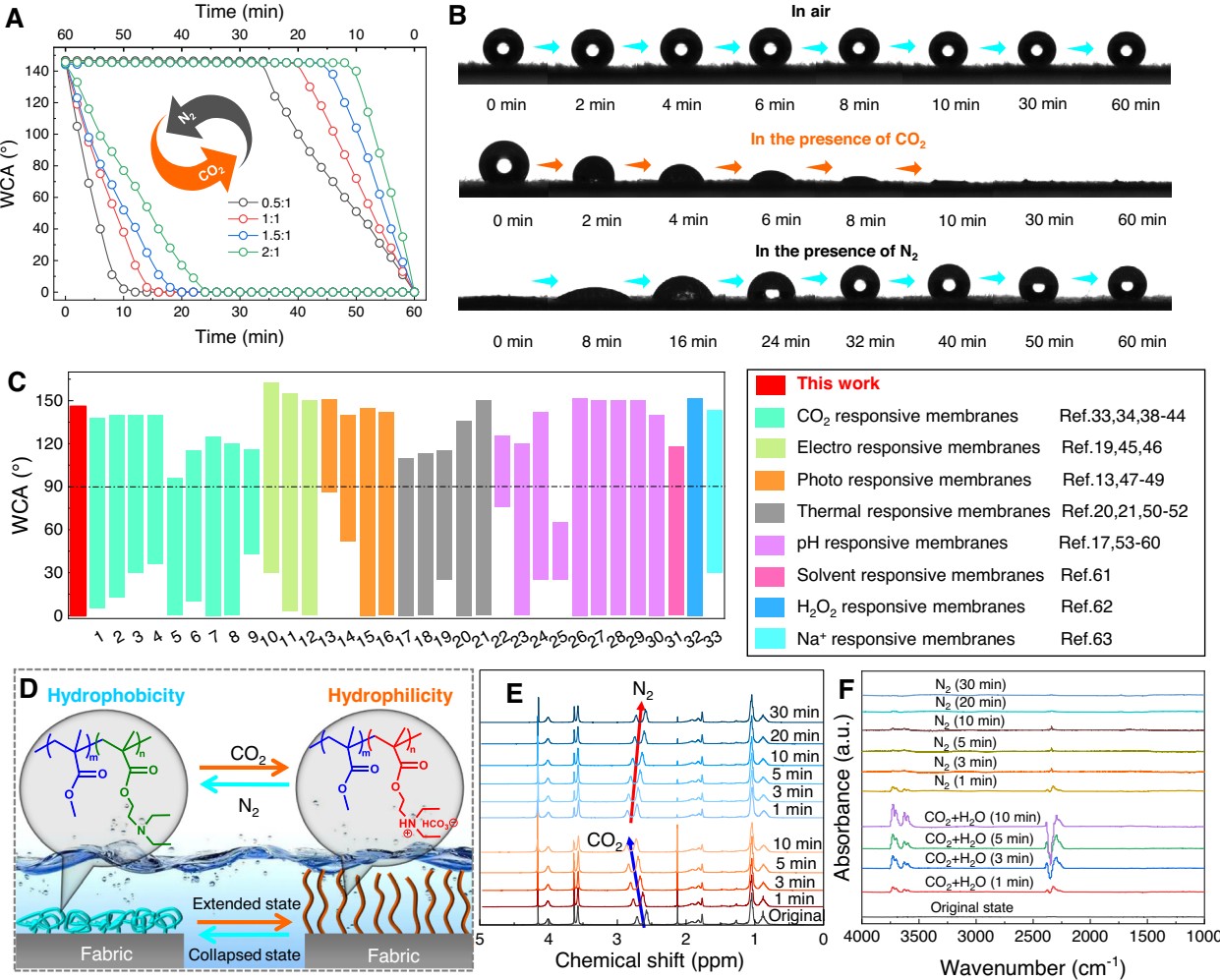

**Fig. 2 | Structure characterization and gas-switchable wettability of PPFM.**
**A** Time-dependent change in WCA for PPFM-0.5 with gap width of 150 μm under $CO_2/N_2$ treatment. The $CO_2$ and $N_2$ were purged at 25 °C. The gas flow rate was 20 mL min⁻¹. **B** Dynamic optical images of the wettability of a water droplet on PPFM-0.5 with gap width of 150 μm in different states. **C** Range of WCA values of our PPFM and other stimuli-responsive membranes reported in the literature[48–73]. **D** Schematic illustration of the surface-wetting mechanism of PPFM under $CO_2/N_2$ stimulation. **E** ¹H NMR spectra of PMMA-co-PDEAEMA copolymer in $D_2O$ and tetrahydrofuran-D8 (1:1) before and after bubbling $CO_2$. **F** In situ FTIR spectra of PPFM-0.5 under $CO_2/N_2$ stimulation.

cross-sectional SEM images (Supplementary Fig. 15) also confirm this by showing that the microfibers are tightly bound by the copolymer phase in the PPFM. Moreover, compared to the spectra of the pristine fabric and PPFM, a new peak appearing at 850 cm⁻¹ belongs to trigonal C-N(-C)-C units in the FT-IR spectrum (Supplementary Fig. 13), and a typical N 1s peak appearing at approximately 286 eV in the XPS spectrum (Supplementary Fig. 14) can be observed in the PPFM, which further verifies that the copolymer is deposited in the PPFM (inside and outside the membrane).

## Gas switchable wettability of the membranes

As the ratio of MMA/DEAEMA in the copolymer plays an important role in the switching ability of the surface wettability of the as-prepared PPFM and the subsequent $CO_2$-tunable emulsion separation performance, a series of studies were carried out to investigate the effect of various MMA/DEAEMA ratios, which was measured by the water contact angle (WCA) under alternating $CO_2/N_2$ stimulation. As shown in Fig. 2A, at the initial state (that is, atmospheric environment), a stable WCA of >140° can be obtained in all PPFMs for 60 min (Fig. 2B), and with the increasing MMA/DEAEMA ratio, these WCAs show slight declination. This result could be explained by decreased DEAEMA

content in the copolymer (Supplementary Fig. 16), which deteriorates the hydrophobicity of the membrane surface. After $CO_2$ treatment, all PPFMs were endowed with a significant transition from high hydrophobicity to superhydrophilicity, with the WCA significantly decreasing from >140° to 0° (Fig. 2A, B). After removing $CO_2$ by passing $N_2$, the wettability of all membranes can return to the initial hydrophobic state. In comparison to previously reported stimuli-responsive membranes (Fig. 2C), PPFM exhibits the largest range of WCAs, revealing good switching surface wettability. Similar phenomena are observed in the underwater oil contact angle (UOCA) and underoil water contact angle (UWCA) of PPFM (Supplementary Fig. 17), in which switching $CO_2/N_2$ bubbling can achieve a reversible wettability transition from underwater superoleophilicity to underwater superoleophobicity. As illustrated in Supplementary Fig. 18, the smart and controllable transition between the two extreme wettability values could be repeated more than 10 times without an obvious fluctuation in the responsiveness, indicating the favorable gas-switching ability of the PPFM. This robust wettability transition can be explained by the protonation and deprotonation effects of amine groups in the PDEAEMA segments of the copolymer (Fig. 2D)[29,33,37]. That is, in neutral aqueous media, the initial PDEAEMA segments are in a dehydrated and chain-collapsed

state, endowing the membrane surface with high hydrophobicity. Upon $CO_2$ treatment, the collapsed PDEAEMA segments are converted into a chain-extended state due to protonation of the tertiary amine groups, making the material favorable for water capture. Subsequent $N_2$ purging causes the stretched PDEAEMA segments to gradually deprotonate, resulting in a coiled conformation and subsequent transition of surface wettability from superhydrophilicity to high hydrophobicity. To confirm this reversible process, the variation of $^1$H NMR spectra of PMMA-*co*-PDEAEMA copolymer and in situ FTIR spectra of PPFMs under $CO_2/N_2$ stimulation were carried out as shown in Fig. 2E, F. Upon $CO_2$ treatment in $D_2O$ and tetrahydrofuran-D8 (1:1), the chemical shifts of PDEAEMA, especially adjacent groups of N atoms (e.g., protons of ethyl groups linked to the tertiary amine group at 2.6 ppm and 2.8 ppm), gradually display a downfield shift (Fig. 2E and Supplementary Fig. 19) as a function of the reaction time; this occurred due to the decreased electron cloud density of the N atom in the amine groups caused by the protonation of amine groups[38]. Upon $N_2$ bubbling, those chemical shifts were recovered to the initial state. The in situ FTIR spectra analysis (Fig. 2F and Supplementary Fig. 20) reveals that the characteristic peaks at 2400–2300 cm$^{-1}$ belong to $CO_2$, and the peaks at 3700–3600 cm$^{-1}$ belong to the -OH groups of $H_2O$, and their intensities gradually increase with increasing $CO_2$ and $H_2O$ penetration time. After $N_2$ is injected, these peaks gradually disappear, confirming that the peaks arise from the reversible absorption of $CO_2$ and $H_2O$.

Furthermore, as shown in Supplementary Fig. 21, the wettability difference of the surface under $CO_2/N_2$ conditions became greater as the MMA/DEAEMA ratio increased. At the MMA/DEAEMA ratio of 0.5, the as-prepared FFPM exhibits excellent gas-responsive change from high hydrophobicity (WCA = 146°) to superhydrophilicity (WCA = 0°) compared with that of other ratios. Together with the pore size analysis (Supplementary Fig. 22), $^1$H NMR (Fig. 2E and Supplementary Fig. 19), EDS line-scan (Supplementary Fig. 8) and in situ FTIR spectra (Fig. 2F and Supplementary Fig. 20) of PPFMs with different MMA/DEAEMA ratios, an optimal MMA/DEAEMA ratio of 0.5 was applied to prepare the PPFM for the following characterizations and performance tests throughout this work.

## Gas switchable immiscible oil/water mixture separation

The gas-tunable surface wettability indicates the feasibility of the as-prepared PPFMs for controllable immiscible oil/water mixture separation. Therefore, a series of proof-of-concept experiments were conducted. Supplementary Fig. 23 exhibits the pure water permeance of the obtained membranes at 25 °C under the stimulation of $CO_2$ and $N_2$ (setup illustration and details in the measurement of water permeance are shown in the ESI). As controls, the water permeance of the pure polyester fabric membrane prepared under the same conditions was also tested. Upon bubbling $CO_2$ through the solution (flow rate: 20 mL/min), the water permeability of the PPFM increased until a plateau was reached after 10 min, while the pure polyester fabric membrane showed no change in water permeability under the same $CO_2$ stimulation. The significant increase in water permeability was due to increased hydrophilicity of the membrane surface (Fig. 2A), decreasing the water intrusion pressure (Supplementary Table 2) and thus facilitating transport of water molecules through the membrane. Afterward, after $N_2$ was passed through the solution (flow rate: 20 mL min$^{-1}$) for 20 min, the membrane surface changed from superhydrophilicity to high hydrophobicity (Fig. 2A), resulting in an increase in the water intrusion pressure (Supplementary Table 2). Therefore, the water permeability gradually recovers to the initial level. Subsequently, various immiscible oil-water mixtures were exploited to investigate the controllable oil/water separation that was driven solely by gravity, as shown in Fig. 3A. Originally, when the mixture was poured onto the membrane treated by water with $CO_2$ bubbling for 10 min, the water colored with methylene blue could rapidly pass

through the membrane, while the oils colored with oil red O could not. Then, the as-prepared membrane was exposed to $N_2$ for 30 min, and the complete opposite transport process could be observed. As shown in Fig. 3B, both the water and oil contents in the corresponding filtrates are below 5 ppm for the four types of oil/water mixture systems, demonstrating unexpectedly high separation efficiencies. This is mainly due to the change in the surface roughness of the membrane under $CO_2/N_2$ stimulation (Fig. 3C, D), in which the presence of $CO_2$ favors the formation of a rougher and more hydrophilic surface and then forms a protection layer; this layer allows the water phase to pass but blocks the oil phase[33,39–41]. Upon exposure to $N_2$, the surface roughness of the membrane recovers to the initial state, thus making the separation process fully reversible.

## Gas switchable emulsion separation

The excellent oil-water separation performance encouraged us to systematically study the potential applications of PPFM in the field of emulsion separation. A sequence of surfactant-stabilized emulsions, including oil-in-water (O/W) and water-in-oil (W/O) types, which are derived from various oils, such as light oils and high-viscosity oils, were prepared as processing targets. The average droplet sizes of all eight kinds of emulsions were measured by a Zetasizer Nano instrument and are shown in Supplementary Fig. 25; these results demonstrate that numerous droplets (-1 μm) are well dispersed in the corresponding emulsions (Supplementary Fig. 26). As shown in Fig. 4A, in contrast to the milky-white state of the original feed emulsions, all filtrates became transparent after separation. Additionally, there is an obvious difference in the droplet size between the feed and the filtrate, as measured by optical microscopy and a Zetasizer Nano instrument (Supplementary Fig. 25). Obvious gas switchable emulsion separation performance can be achieved by PPFM-0.5, in which $CO_2$ stimulation allows the membrane to separate W/O emulsions, while the $N_2$-stimulated membrane favors the separation of O/W emulsions. A moisture meter and total organic carbon (TOC) analyzer were used to analyze the purified water and oils, respectively. As shown in Fig. 4B, C, the contents of the emulsion in the filtrates after separation are below 40 ppm, with a purity above 99.5% for the W/O emulsions, and below 20 ppm, with a purity above 99.5% for the O/W emulsions. Especially for *n*-hexane-based emulsions, the contents of the emulsion in the filtrates are as low as 19 ppm (O/W) and 12 ppm (O/W), respectively, which are far less than that for other reported membranes (Supplementary Fig. 27); these results demonstrate the superior separation performance of these stable emulsions, which is observed regardless of the oil-in-water or water-in-oil type. In addition, regardless of the water-removing or oil-removing mode, the separation efficiency of all emulsions is above 99.60% and no obvious changes are observed over 20 cycles (Supplementary Fig. 28); thus, the emulsions possess excellent reusability for emulsion separation.

To clarify the underlying mechanism of gas switchable emulsion separation of PPFM-0.5, a series of density functional theory (DFT) calculations were performed. As demonstrated by previous reports and this work (Fig. 2D), the amine groups in DEAEMA can be protonated and deprotonated through $CO_2/N_2$ treatment. The protonation processes of DEAEMA were first calculated. Clearly, as shown in Supplementary Fig. 29, it is easy to protonate DEAEMA with $H_3O^+$ (denoted as DEAEMA+$H_3O$). Then, the influence of $H_3O^+$-protonated DEAEMA on water/oil adsorption was also studied. As shown in Supplementary Fig. 30, the adsorption energies ($E_{ad}$) of water and selected oils on pristine and protonated DEAEMA are calculated. Before protonation, the $E_{ad}$ of water on DEAEMA is much lower than that of the oils on it, indicating strong interactions between the oils and DEAEMA. After $CO_2$ treatment, the $E_{ad}$ is reversed for water and oils, which fully agrees with the following experimental results: a much higher WCA, smaller UOCA and ultralow adhesion force to water droplets in oil can be obtained in the original PPFM-0.5, while a smaller WCA, higher UOCA and low

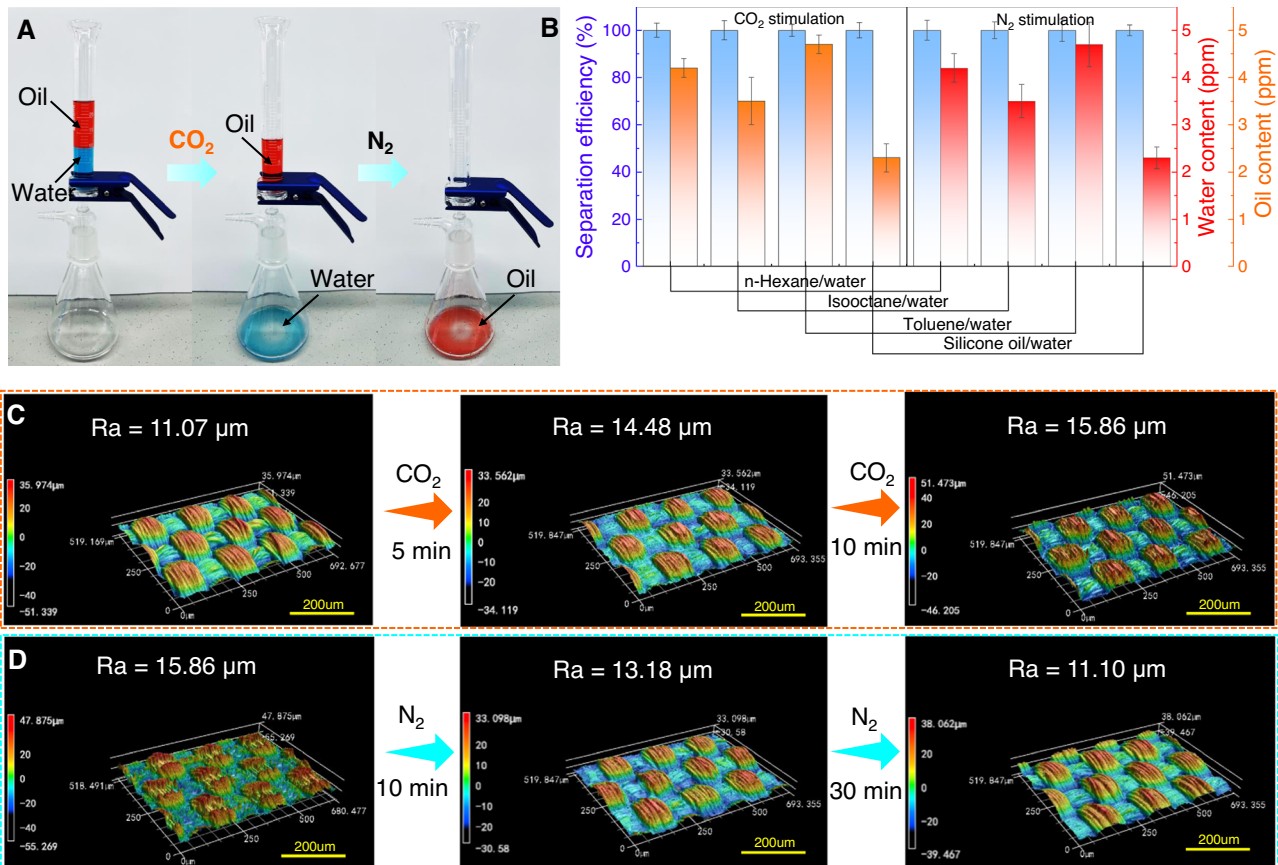

**Fig. 3 | Gas switchable immiscible oil/water mixture separation. A** Process of separating the immiscible oil/water mixture upon $CO_2/N_2$ stimulation at 25 °C. **B** Separation efficiency and oil or water content of PPFM-0.5 with gap width of 150 μm for four types of oil/water mixture systems under $CO_2/N_2$ stimulation. **C, D** Variation in the surface roughness of PPFM-0.5 under $CO_2/N_2$ stimulation. The error bars represent the standard deviation and were calculated on the basis of at least three data points measured from different samples.

adhesion force to oil droplets in water can be obtained in the $CO_2$-treated PPFM-0.5 (Figs. 2D and 4D and Supplementary Fig. 17 and 32). For such reversion, the interactions between DEAEMA ($DEAEAM+H_3O^+$) and water/oils were deeply analyzed. Clearly, as shown in Fig. 4E and Supplementary Fig. 31, the van der Waals force dominated the adsorption of water (green areas) before protonation, and the H-bond dominated the adsorption after protonation. However, the adsorption of oils was also dominated by van der Waals forces before protonation, and the rapidly increasing repulsive force (green to red areas) should heavily prevent oils from closing to $DEAEMA+H_3O^+$. As a result, the oil adsorption is weaker than that of water, leading to a different oil/water mixture separation during experimental work.

In fact, the components in real effluents discharged from industrial processes and daily life are rather complicated. Multiphase emulsion mixtures and various pollutants usually coexist together[42,43], and evaluations on the continuous separation performance of PPFM-0.5 for the aforementioned complicated emulsion systems are urgently needed. In this work, n-hexane, silicone oil, toluene, and isooctane-based emulsions were selected as model multiphase emulsions. The operating principle is displayed in Fig. 5A and Supplementary Fig. 33. PPFM-0.5 first contacts the O/W phase emulsion and allows the filtrates (i.e., water) to enter the water-containing region under $CO_2$ stimulation. Thereafter, PPFM-0.5 contacts the W/O phase emulsion. Following the removal of $CO_2$ by $N_2$ bubbling, PPFM-0.5 allows the filtrates (i.e., n-hexane or silicone oil) to enter the oil-containing region. The separation efficiency of each process can be stabilized at >99.5% (Fig. 5B and Supplementary Fig. 34). Therefore, by altering the $CO_2/N_2$ stimulation, PPFM-0.5 can achieve efficient continuous separation of

multiphase emulsion mixtures. The industrial application potential of PPFM-0.5 was also explored by studying its scalability and mechanical stability. As shown in Fig. 5C–E, a large-scale PPFM-0.5 with an area of 3600 $cm^2$ can be successfully produced and presents stable separation performance. Supplementary Fig. 35 shows the mechanical stability of PPFM-0.5, as evaluated by tape-peeling, abrasion, ultrasonic peeling and scratch tests. As seen, the separation performance of PPFM-0.5 shows a slight fluctuation after the multicycling mechanical tests. The SEM images of the membrane after the mechanical tests are displayed in Supplementary Figs. 36–39, showing that the membrane surface morphology can remain the same after the mechanical test. The results demonstrate that PPFM-0.5 exhibits excellent mechanical stability and reusability, resisting wear and tear in practical applications.

### Gas-controlled self-cleaning performance

In practical applications, membrane fouling can deteriorate the separation performance and shorten the lifetime of membranes[44–47]. Due to the robust switchable wettability of the membrane surface under $CO_2/N_2$ stimulation, the PPFM-0.5 shows promise for application in the self-cleaning field. As a proof of concept, the self-cleaning performance of PPFM-0.5 was assessed using n-hexane-based O/W emulsions containing BSA and TC and n-hexane-based W/O emulsions as model organic foulants. Lab-scale dead end filtration equipment was used to monitor the variation in membrane flux during five cycles of the filtration process under gravity conditions. As illustrated in Fig. 5F, the membrane flux abruptly decreases from 325 $Lm^{-2}h^{-1}$ to 100 $Lm^{-2}h^{-1}$ during one cycle in the case of O/W

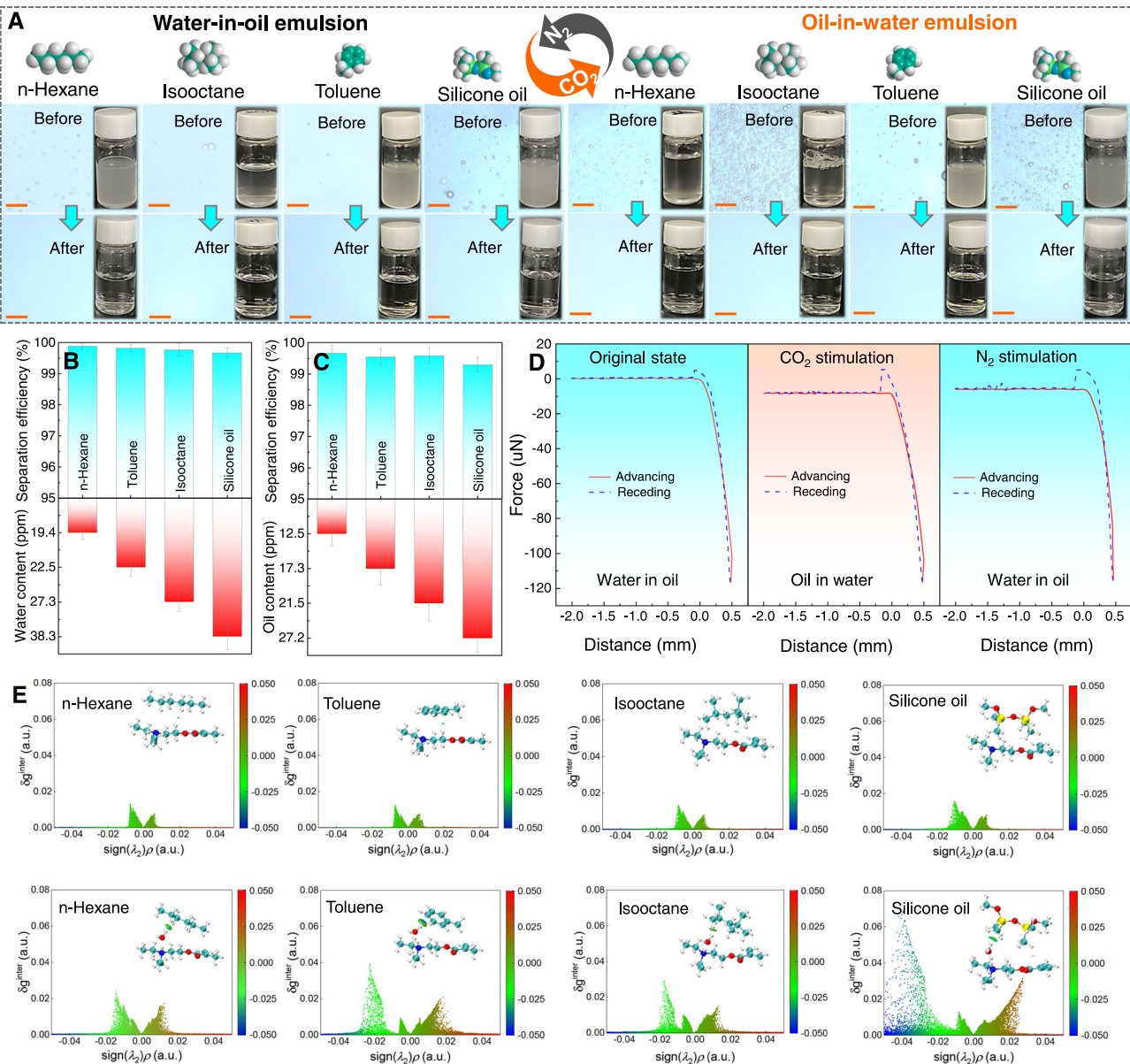

**Fig. 4 | Gas switchable emulsion separation. A** Photographs of various O/W and W/O emulsions under $CO_2/N_2$ stimulation and the corresponding optical microscopic images. The scale bar is 20 μm. **B, C** Separation efficiency and oil or water content of PPFM-0.5 with gap width of 150 μm for the W/O (**B**) and O/W (**C**) mixture systems at 25 °C under $CO_2/N_2$ stimulation. **D** Adhesion force-distance curves of the underoil water or underwater oil droplet on the PPFM-0.5 surface with gap width of 150 μm under $CO_2/N_2$ stimulation. **E** 2D scatter plot of noncovalent interactions between adsorbed species and DEAEMA (top)/protonated DEAEMA (down). The error bars represent the standard deviation and were calculated on the basis of at least three data points measured from different samples.

emulsions containing BSA and TC, and obvious residues can be detected on the membrane surface, indicating serious membrane fouling (Supplementary Figs. 40–43). After washing with $CO_2$ bubbling for 15 min, the membrane flux can be effectively restored, and the membrane surface becomes rather clean, indicating that the BSA and TC fouling on the membrane surface can be efficiently removed (Supplementary Figs. 40–43). For the *n*-hexane-based W/O emulsions, PPFM-0.5 exhibits similar cyclic stability (i.e., first decreases and then completely recovers after $CO_2$ bubbling). The working mechanism for self-cleaning involves the switchable nature transition in hydrophilicity/hydrophobicity on the membrane surface under $CO_2$ stimulation, which is governed by the protonation and deprotonation of the tertiary amine groups in PDEAEMA (Fig. 5G). More importantly, the cleaning efficiency of PPFM-0.5 for all emulsion systems is up to 99.5%, indicating that PPFM-0.5 possesses excellent

self-cleaning performance toward various types of emulsion systems with various foulants.

In summary, we have proposed a CFCS strategy to develop a scale $CO_2$-responsive membrane with gas-tunable surface wettability, which shows robust switching between hydrophobicity/superoleophilicity and superhydrophilicity/oleophobicity under alternating $CO_2/N_2$ stimulation. The membrane can be applied to various oil-in-water and water-in-oil emulsions and multiphase emulsion mixtures due to its switchable transport property and demonstrates high separation efficiency, permeability, recyclability, and self-cleaning performances. Moreover, the CFCS strategy features a simple preparation process, which is conducive to large-scale production. We envision that both the membrane design and surface property control strategies will provide a theoretical and technological reference for the fabrication of smart membranes for numerous practical applications, such as

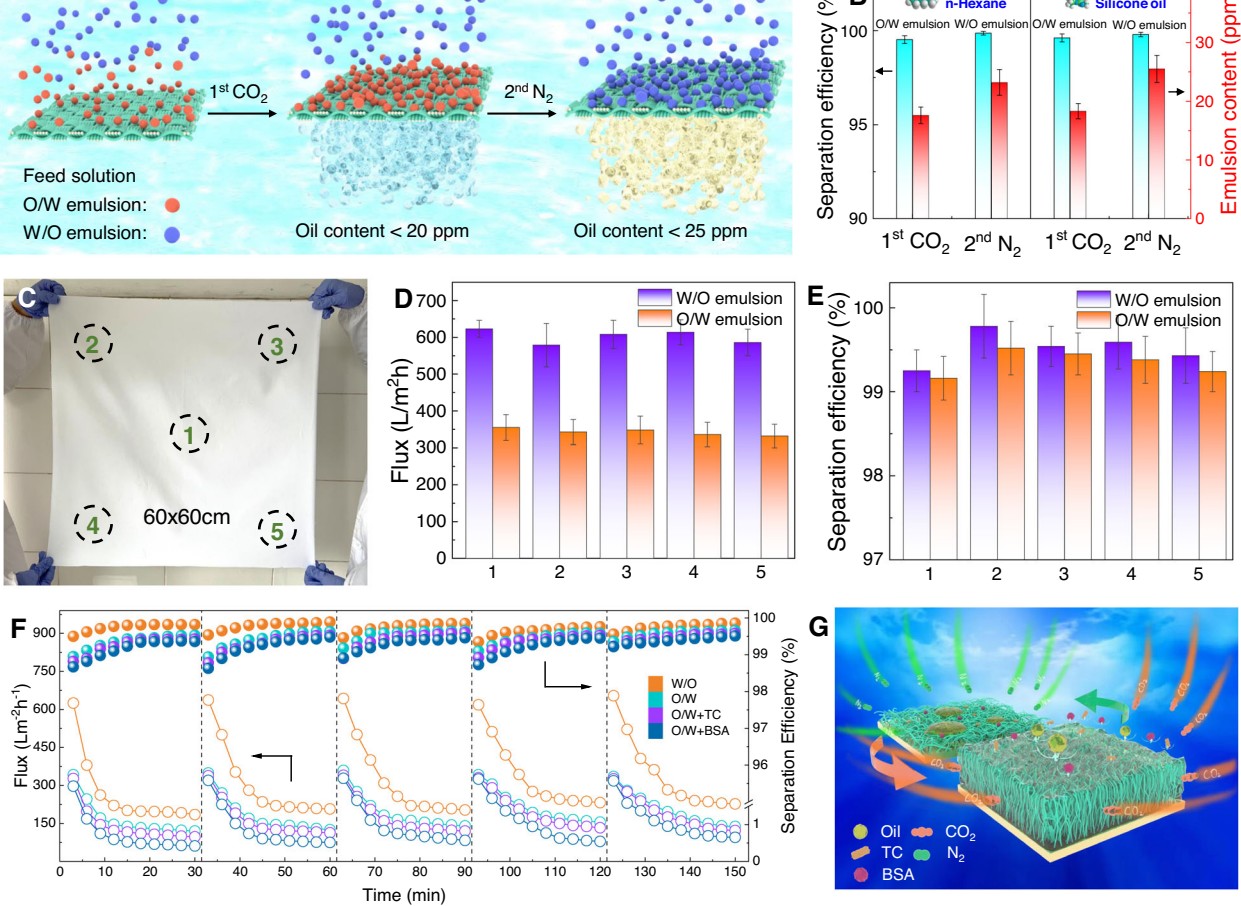

**Fig. 5 | Gas switchable multiphase emulsion mixtures separation and self-cleaning performance. A** Schematic showing the two-step separation process for the multiphase emulsion system under $CO_2/N_2$ stimulation. **B** Separation efficiency of PPFM-0.5 with gap width of 150 μm in the 1st and 2nd step operations. The emulsion content is oil content for O/W emulsion and water content for W/O emulsion, respectively. **C** Photo of the prepared large-size PPFM with 60 × 60 cm. **D**, **E** Separation efficiency and flux of the selected area in PPFM with gap width of 150 μm for both W/O and O/W emulsions at 25 °C. The W/O and O/W emulsions used are water/n-hexane and n-hexane/water emulsions, respectively. **F** Self-cleaning performance of PPFM-0.5 with gap width of 150 μm for both W/O and O/W emulsions with or without organic foulants at 25 °C. The W/O and O/W emulsions used are water/n-hexane and n-hexane/water emulsions, respectively. The concentrations of BSA protein and TC in the O/W emulsion were 1 g/L and 50 mg/L, respectively. **G** Schematic illustration of the gas-controlled self-cleaning mechanism. The error bars represent the standard deviation and were calculated on the basis of at least three data points measured from different samples.

---

effluent treatment, spill oil cleanup, and separation of commercially relevant emulsions.

## Data availability

The source data underlying Figs. 2–5 and Supplementary Figs. 6–8, 12, 13, 15–22, 24–27 and 33 are provided as a Source Data file. Source data are provided with this paper.

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

## Acknowledgements

This work was financially supported by the National Key Research and Development Program of China (2021YFB3802600 (L.D.), 2021YFA1101300 (X.Y.)), the Fundamental Research Funds for the Central Universities (JUSRP622035) (L.D.), the Natural Science Foundation of Xinjiang Uygur Autonomous Region (2022D01D030) (L.D.).

## Author contributions

Y.W., Y.B., C.Z., and L.D. conceived and designed the experiments. Y.B., C.Z., and L.D. supervised the study and experiments. Y.W. conducted the membrane fabrication, characterization, and performance tests. S.L., J.L., B.Z., and L.D. supported all the characterizations. Y.W., Y.B., C.Z., S.L., J.L., B.Z., and L.D. analyzed the experimental results. S.Y. and D.R. conducted the DFT simulation. J.Z., Z.C., and J.X. conducted the COMOSL simulation. Y.W., S.Y., and J.Z. wrote the paper. X.Y. and L.D. revised the manuscript. All the authors discussed the results and provided comments.

## Competing interests

The authors declare no competing interests.
