## [Peer Review File · Nature Communications]

Scalable and switchable CO₂-responsive membranes with high wettability for separation of various oil/water systemsReviewers' Comments:

Reviewer #1:

Remarks to the Author:

The manuscript reported the development of a CO₂ responsive membrane based on the stimulate-responsive copolymer. The tunable surface wettability and emulsion separation efficiency have been investigated. However, there is much work on CO₂-switchable membranes reported for controllable separation of oil/water mixtures based on the same kind of polymer (Angew 2015, 54, 8934; Langmuir 2017, 33, 43, 11936; ACS Appl. Mater. Inter. 2019,11,9, 9367). Considering the novelty, the article belongs to a special area that may fit the other journal.

1. The author indicated in line 145 that "PMMA segments provide a sufficient stability of copolymers in water". As the PMMA is not water-soluble, so the this statement may not proper here. The author should provide the explanation that why choosing PMMA as one segment.
2. Whether the PMMA-PDMAEMA segment ratio would influence the spreading ability of the polymer on the acrylic plate? Thus influence the final effect of the membrane.
3. The thickness of the film on both side should be provided.
4. The author mentioned that the water contact angle change of the membrane is mainly due to the roughness change of the surface. Please provide the roughness measurement of the surface to support this discussion. Whether the functional groups exposed outside the surface changes would influence by the deposition condition, thus have an impact on the hydrophobicity of the surface?
5. How about the stability of the membrane under exposure to organic solvent, like THF?
6. The authors indicated that PPFM-0.5 was applied to prepare the membrane. The response time of the membrane is still quite long (10min and 25min respectively). How about further increase the DEAEEMA segment ratio?
7. Some typo need to revise, e.g. Page4, line 141, "CO₂-resonsive" should be "responsive"

Reviewer #2:

Remarks to the Author:

Many researchers have tried to make readily reversible membranes that can cleanly and reversibly interconvert between selectively water-permeable and selectively oil-permeable. That has been achieved to some extent in various recent reports but none as cleanly as those described here. In addition, a goal in membrane work has been to develop membranes that can be cleaned with merely water or carbonated water without the use of harsh chemicals. This manuscript presents membranes that simultaneously achieve both of these goals. The authors also demonstrate that the membranes can be used at least 10 times and that they can be easily and cheaply prepared. I'm very impressed by the work and congratulate the authors on their achievement. The paper should be accepted with only minor changes, as described below.

Minor points

- line 148: a copolymer solution is mentioned but the solvent should have been stated.
- lines 423 and 426: "CO₂ aeration" is not clear for two reasons. First, "aeration" means addition of air. CO₂ is not air. Second, I suspect that the authors washed the membrane with carbonated water, not just CO₂. However, the presence of water in this membrane washing process isn't mentioned. It should be.
- In Figure S19 the graph shows that water passes through the membrane even after the membrane has been treated with N₂. However, Table S2 suggests that the water intrusion pressure after N₂ treatment is 17 kPa. That seems contradictory. Why did water pass through after N₂ treatment if the intrusion pressure was greater than 0 kPa? Please explain this.
- The captions in the figures such as Fig S16 don't mention what gap was used to prepare the membranes. All figure and tables giving data about membranes should specify enough data so that

the reader can understand what membrane was used. This data should include the gap used.

Most of the issues are to do with the English language.

- line 43 "inconspicuous" should be "inadequate"
- many places: "resonsive" should be "responsive"
- line 113: "enlargement" should be "scale-up"
- lines 125 and 150: "inner" should be "inside"
- the paragraph from lines 329 to 376 is far too long. Break it into two paragraphs after "reusability" on line 353.
- SI line 228. What does "wathet" mean? I don't think that's a word in the English language.

Reviewer #3:

Remarks to the Author:

Smart membranes have been reported broadly, but most of the membranes are responsive toward stimuli such as pH, salt, and humidity, and temperature, etc. There are relatively less reports in gas-induced responsiveness, which makes this work an interesting article focusing on CO₂-responsive membrane for separating various oil/water systems. Exploration of the CO₂-responsive polymers into membrane separation field is indeed very attractive. Considering the large membrane size and efficiency in tuning surface properties of their membranes, the method developed in this work for fabricating the CO₂-resonsive membrane is impressive. Under CO₂/N₂ stimulation, the membrane shows excellent switching wettability in comparison to other reported membranes, resulting in outstanding separation efficiency (>99.9%) for various oil/water systems and self-cleaning performance. The structural and chemical properties of the prepared membranes are thoroughly characterized, and the reviewer appreciate that the results is well backed up by theoretical simulations. Overall, I am in favor of its publication after addressing the following comments:

(1) In the introduction on CO₂-responsive polymers, the list of references seems incomplete and the important contributions of groups like the P. Theato group (University of Hamburg) and the Q. Yan (Fundan University) are not mentioned. This should be extended.

(2) What's the operating temperature for separation of different oil/water systems? Because the temperature can also impact on the deprotonation process of amine groups of PDEAEMA.

(3) Authors claim that they exploited capillary force-driven confinement self-assembling (CFCS) strategy to prepare a scalable and stable CO₂-resonsive membrane for smart separation of various oil/water systems. While the mechanism is intriguing, it is not fully clear to me how to define confinement, as this is highly related to the dimension of pore dimensions. Authors would want to elaborate with more details in this regard.

(4) Authors should add the error bars for all membrane separation measurements, such as that in Fig. 5b, 5e, etc.

(5) The two-step separation process for multiphase emulsions system is interesting, but author just test two emulsions. Other emulsions used in this work should be also tested. Moreover, why not simply put all data (separation efficiency in Fig. 5B and emulsion content in Fig. S29) in the same panel. This would easily allow readers to grasp what's going on.

(6) As author state, CO₂ is directly bubbled into emulsion solution during test. How about the effect of CO₂ bubbling on stability of emulsion? This is should be investigated.

(7) In the Fig. 3 C, D, there is lack of scale bar. Quality of Fig. 31-35 in SI are needed to be improved, the scale bars are difficult to see clearly.

(8) Language and expression of this work should be further improved.

Response to comments

Reviewer #1

General Comments:

The manuscript reported the development of a CO₂ responsive membrane based on the stimulate-responsive copolymer. The tunable surface wettability and emulsion separation efficiency have been investigated. However, there is much work on CO₂-switchable membranes reported for controllable separation of oil/water mixtures based on the same kind of polymer (Angew 2015, 54, 8934; Langmuir 2017, 33, 43, 11936; ACS Appl. Mater. Inter. 2019,11,9, 9367). Considering the novelty, the article belongs to a special area that may fit the other journal.

Response:

First of all, we appreciate the reviewer's positive comments on this work. The main reason why the reviewer questions our manuscript is that there are some similar works having been reported. However, the reported publications are different from our manuscript in terms of membrane preparation, membrane structure and separation system. For example, the references mentioned by this reviewer, as follows:

1) Angew. Chem., Int. Ed., 2015, 54, 8934.

This work reported a CO₂-responsive nanofibrous membrane via electrospinning technique for oil-water separation. Although the similar polymer was used in this work, the resulted membranes show a large pore size and wide pore size distribution because they are formed by random stacking of nanofibers. Therefore, these membranes can be used for immiscible oil/water mixtures, not for emulsified oil-water mixtures. Moreover, their separation efficiencies (e.g., water contents in hexane and n-heptane were ~30 and 60 ppm, respectively) are lower than those of our membranes (e.g., water contents in the corresponding filtrates are below 5 ppm).

2) Langmuir 2017, 33, 11936.

This work reported a CO₂-responsive membrane via high internal phase emulsion for oil/water separation. Although the resulted membranes show relatively uniform pore size distribution, their pore sizes are very large (>3 μm). Therefore, these membranes can not be used for separation of emulsified oil-water mixtures. Moreover, this method is limited to laboratory and difficult to industrialize.

3) ACS Appl. Mater. Inter. 2019,11, 9367.

This work reported a CO₂-responsive aerogel for separation of surfactant-stabilized emulsions. Although the aerogel shows good switchable oil-water separation performance, the aerogel is not essentially categorized as separation membrane. Moreover, this work still used chemical grafting method to prepare CO₂-responsive aerogel, which is not only difficult to industrialize, but also lead to inhomogeneous presence of responsive moieties the substrate. Therefore, compared to our membrane, the separation efficiencies of aerogel are still lower (the water and oil contents of the corresponding filtrates were above 20 ppm).

Based on above discussion, we restate the novelty of our manuscript as follows: i) robust switching wettability under CO₂/N₂ stimulation, resulting in high separation

efficiency (>99.9%) for all types of oil/water systems (e.g., immiscible oil/water mixtures, surfactants-stabilized emulsions and multiphase emulsions), ii) large-scale production, the proposed method in this work generates the membrane with large area up to 3600 cm², iii) excellent self-cleaning performance (the cleaning efficiency of 99.5%) toward various types of emulsion systems containing various foulants.

Specific comments:

Comment 1:

The author indicated in line 145 that “PMMA segments provide a sufficient stability of copolymers in water”. As the PMMA is not water-soluble, so this statement may not proper here. The author should provide the explanation that why choosing PMMA as one segment.

Response 1:

Thanks for your comment. Because the PDEAEMA is hydrophilic and water-soluble under CO₂ stimulation, the formed polymer coating will be unstable and easy to fall off from the fabric substrate if pure PDEAEMA is chosen as CO₂-responsive polymer. Because PMMA is insoluble in water and no CO₂ response, the presence of PMMA fragments can keep the copolymer insoluble in water under CO₂ stimulation, thus preventing the copolymer coating from falling off and enhancing the stability of copolymers in water. To avoid the misunderstanding, we have corrected the description in the revised manuscript as follows:

“A series of PMMA-co-PDEAEMA copolymers synthesized by radical copolymerization reactions were chosen as CO₂-responsive polymers in this work, in which the PDEAEMA segments endow the copolymers with high sensitivity to CO₂, while the PMMA segments that are insoluble in water, prevent falling off of copolymers from fabric substrate under CO₂ stimulation.”

Comment 2:

Whether the PMMA-PDMAEMA segment ratio would influence the spreading ability of the polymer on the acrylic plate? Thus influence the final effect of the membrane.

Response 2:

Thanks for your comment. We have measured the dynamic CA transformation of polymer solution with different MMA/DEAEMA segment ratios on surface of acrylic plate. As shown in Fig. S1, all copolymer solutions can spread out completely in one second, confirming that the MMA/DEAEMA segment ratio has little effect on the spreading ability of the synthesized polymers on the acrylic plate. This is mainly due to the combined results of good solubility of all PMMA-PDMAEMA copolymers in THF and low copolymer concentration (10 wt%). To facilitate the understanding, we have added the Fig. S1 in the revised SI.

1:1	27.5°	18.8°	12.3°	6.4°	3.1°	0°
	0s	0.2s	0.4s	0.6°	0.8°	1°
1.5:1	26.3°	20.6°	16.3°	10.5°	4.3°	0°
	0s	0.2s	0.4s	0.6°	0.8°	1°
2:1	26.4°	21.2°	16.3°	9.4°	5.2°	0°
	0s	0.2s	0.4s	0.6°	0.8°	1°

Figure S1. Dynamic CA transformation of PMMA-co-PDEAEMA copolymer droplet with different MMA/DEAEMA ratios on the surface of the acrylic plate over time.

Comment 3:

The thickness of the film on both sides should be provided.

Response 3:

Thanks for your good comment. We used EDS line scan to measure the thickness of the membrane. Because the PMMA-PDMAEMA copolymer has similar C, O and H elements to fabric substrate, it is difficult to directly measure the polymer coating thickness based on these elements. In order to solve this problem, we used FeCl₂ as tracer agent. We added 50 mg/mL FeCl₂ into polymer solution during the membrane preparation process. Because the Fe element can be easily detected through EDS line scan, the polymer coating thickness can be measured based on the distribution of Fe element along the cross section of membrane. As shown in Fig. S6, with the increase of gap width, the thickness difference between two sides gradually becomes larger. At the gap width of 150 μm, both A side and B side have the same coating thickness of 6 μm. In contrast, under the fixed gap with (e.g., 150 μm), the coating thickness of both sides is almost unchanged with the increase of segment ratio (Fig. S7). These results are consistent with the SEM images (Fig. 1B, S3 and S11).

Based on above discussion, we have added figures of EDS line scan and corresponding discussion in the revised manuscript and SI as follows:

In the revised manuscript: “Cross-sectional SEM images and EDS line-scan (Figs. S4 and S6) further confirm this morphology difference, as the thickness of the copolymer layer on the A side shows a significant increase, while that on the B side only shows a slight increase with increasing gap width (Fig. S6).”

In the revised SI: “The copolymer coating thickness of membranes was measured by the EDS line scan. Since the PMMA-PDMAEMA copolymer has similar C, O and H elements to fabric substrate, it is difficult to directly measure the coating thickness based on these elements. In order to solve this problem, we added 50 mg/mL FeCl₂ as tracer agent into polymer solution during the membrane preparation. Because the Fe element can be easily detected through EDS line scan, the coating thickness can be measured based on the distribution of Fe element along the cross section of membranes.”

Figure S6. EDS line-scan element profile of as-prepared PPFMs (MMA/DEAEMA ratio of 0.5) with different gap widths. (A) 150 μm , (B) 200 μm , (C) 250 μm , (D) 300 μm . Fe element is used as tracer agent for measuring the copolymer coating thickness of the membrane on both sides.

Figure S7. EDS line-scan element profile of as-prepared PPFMs with different MMA/DEAEMA ratios. (A) 0.5:1, (B) 1:1, (C) 1.5:1 and (D) 2:1. Fe element is used as tracer agent for measuring the copolymer coating thickness of the membrane on both sides. The gap width used is 150 μm .

Comment 4:

The author mentioned that the water contact angle change of the membrane is mainly due to the roughness change of the surface. Please provide the roughness measurement of the surface to support this discussion. Whether the functional groups exposed outside the surface changes would influence by the deposition condition, thus have an impact on the hydrophobicity of the surface?

Response 4:

Thanks for your comment. In this work, the roughness of membrane surface was measured by the 3D laser microscopical imaging system (VK-150K, KEYENCE, Japan), and the results are shown in Figure 3 C, D. Moreover, according to reviewer's comment, we used ATR-FTIR spectra to investigate the effect of gap width on change of the functional groups exposed outside the surface, because the ATR-FTIR spectra can provide information on functional groups near the surface ($\sim 1 \mu\text{m}$) of an internal reflection element (IRE) (A.J. Margenot, F.J. Calderón, K.W. Goyne, et al, IR Spectroscopy, Soil Analysis Applications, *Encyclopedia of Spectroscopy and Spectrometry (Third Edition)*, 2017, 448-454.). As shown in Fig. R1, peaks appearing at 725 cm^{-1} and 1145 cm^{-1} belong to benzene ring groups of the fabric and C-N groups of the PMMA-co-PDEAEMA copolymer, respectively. Therefore, the peak intensity ratio of I1145/I725 can be used to compare the relative change in functional groups near the surface. With increasing the

gap width, the peak intensity ratio is almost unchanged, confirming that the deposition condition has no influence on the change of functional groups outside the surface.

Fig. R1. ATR-FTIR spectra and peak intensity ratio at 725 and 1145 cm⁻¹ of PPFM-0.5 with different gap widths.

Comment 5:

How about the stability of the membrane under exposure to organic solvent, like THF?

Response 5:

Thanks for your comment. Because the copolymer can be dissolved in THF, the polymer coating would be dissolved, destroying the stability of the membrane in THF.

Comment 6:

The authors indicated that PPFM-0.5 was applied to prepare the membrane. The response time of the membrane is still quite long (10 min and 25 min respectively). How about further increase the DEAEMA segment ratio?

Response 6:

Thanks for your comment. It is well known that the CO₂-switching process is composed of two steps: (1) diffusion of the gas to the switchable group and (2) the reaction to enact the switch. Because CO₂ has moderate solubility in water, the diffusion of CO₂ is slow, retarding the reaction of CO₂ with the switchable group and then increasing the response time. Because of much lower solubility of N₂ than CO₂ in water, the diffusion of N₂ is very slow, which means that much longer time is needed for N₂ to remove CO₂ from water during the deprotonation steps of switchable group. It is for above reason that the CO₂-responsive membranes generally have a long response time. Actually, the response time of our membrane is the same as many literatures (*RSC Adv.*, 2015, 5, 35622; *ACS Appl. Mater. Interfaces* 2021, 13, 2694; *Chem. Commun.*, 2013, 49, 8377.), but faster than others (*Langmuir*, 2017, 33, 11936; *RSC Adv.*, 2017, 7, 39465; *ACS Macro Lett.*, 2018, 7, 431; *ACS Appl. Mater. Interfaces* 2020, 12, 24363.). Moreover, we have also investigated the effect of higher DEAEMA segment ratio on the response time of membrane according to reviewer's comment. As shown in Fig. R2,

the change rate of contact angle under higher DEAEMA segment ratios (i.e., 0.3:1 and 0.1:1) is almost the same as the membrane under DEAEMA segment ratio of 0.5:1 (i.e., PPFM-0.5). This confirms that further increasing the DEAEMA segment ratio hardly changes the response time of membrane.

Fig. R2 Time-dependent change in WCA for PPFM-0.5 with gap width of 150 μm under CO_2/N_2 treatment.

Comment 7:

Some typo needs to revise, e.g., Page4, line 141, “ CO_2 -resonsive” should be “responsive”

Response 7:

Thanks for your comment. We apologized for our mistakes. We have corrected them in the revised manuscript. Moreover, the language of the manuscript has been edited by a professional technical editing service. The certificate of professional editing is listed as follows:

This document certifies that the manuscript

Scalable and unprecedentedly wettability-switchable CO₂-responsive membrane for smart separation of various oil/water systems

prepared by the authors

Yangyang Wang, Shaokang Yang, Jingwei Zhang, Zhuo Chen, Bo Zhu, Jian Li, Shijing Liang, Yunxiang Bai, Jianhong Xu, Dewei Rao, Liangliang Dong*, Chunfang Zhang, Xiaowei Yang

was edited for proper English language, grammar, punctuation, spelling, and overall style by one or more of the highly qualified native English speaking editors at SNAS.

This certificate was issued on **January 13, 2023** and may be verified on the SNAS website using the verification code **2BD7-5036-8534-C9F4-E33B**.

Neither the research content nor the authors' intentions were altered in any way during the editing process. Documents receiving this certification should be English-ready for publication; however, the author has the ability to accept or reject our suggestions and changes. To verify the final

SNAS edited version, please visit our verification page at secure.authorservices.springernature.com/certificate/verify.

If you have any questions or concerns about this edited document, please contact SNAS at support@as.springernature.com.

Reviewer #2:

General Comments:

Many researchers have tried to make readily reversible membranes that can cleanly and reversibly interconvert between selectively water-permeable and selectively oil-permeable. That has been achieved to some extent in various recent reports but none as cleanly as those described here. In addition, a goal in membrane work has been to develop membranes that can be cleaned with merely water or carbonated water without the use of harsh chemicals. This manuscript presents membranes that simultaneously achieve both of these goals. The authors also demonstrate that the membranes can be used at least 10 times and that they can be easily and cheaply prepared. I'm very impressed by the work and congratulate the authors on their achievement. The paper should be accepted with only minor changes, as described below.

Response:

Many thanks for the reviewer's positive comments and useful suggestions. According to the reviewers' suggestions, some necessary discussions were added in the revised paper. Meanwhile, some of the wording and language errors were also corrected.

Specific comments:

Comment 1:

- line 148: a copolymer solution is mentioned but the solvent should have been stated.

Response 1:

Thanks for your comment, we have added the solvent in the revised manuscript as follows:

“After that, 10 wt% of the copolymer solution (THF as solvent) was slowly injected into the edge of the gap.”

Comment 2:

• lines 423 and 426: “CO₂ aeration” is not clear for two reasons. First, “aeration” means addition of air. CO₂ is not air. Second, I suspect that the authors washed the membrane with carbonated water, not just CO₂. However, the presence of water in this membrane washing process isn't mentioned. It should be.

Response 2:

Thanks for your useful suggestion. The formulation of “CO₂ aeration” is indeed not accurate enough. We have replaced "CO₂ aeration" with "CO₂ bubbling" in the revised manuscript. In addition, the membrane washing process was indeed used by the carbonated water. In order to make it clear, we have provided detailed description of membrane washing process in the revised manuscript and SI as follows:

In the revised Manuscript: “After washing with CO₂ bubbling for 15 min, the membrane flux can be effectively restored, and the membrane surface becomes rather clean...”

In the revised SI: “After filtration of the emulsion, the membrane was immersed in water with bubbling CO₂ for 15 min and the next cycle of filtration was then conducted.”

Comment 3:

• In Figure S19 the graph shows that water passes through the membrane even after the membrane has been treated with N₂. However, Table S2 suggests that the water intrusion pressure after N₂ treatment is 17 kPa. That seems contradictory. Why did water pass through after N₂ treatment if the intrusion pressure was greater than 0 kPa? Please explain this.

Response 3:

Thanks for your good comment. We apologized for our unclear descriptions about the water and oil intrusion pressures of membrane. Actually, in the Table S2, the intrusion pressure of membrane in presence of N₂ is measured based on completely dried membrane surface by continuously purging surface with N₂ for a long time. In order to be consistent with the water permeability experiment, we measured the intrusion pressures of membrane under N₂ bubbling time of 20 min. As shown in the revised Table S2, after bubbling N₂ for 20 min, all the oil intrusion pressures of membrane are 0 kPa, which are the same as the results of completely dried membrane. But the water intrusion pressure decreases from 17 kPa to 6 kPa. This is mainly because CO₂ is not completely removed from water, keeping some hydrophilic and protonated groups still existing in the membrane. Because the drive force in water permeability experiment is 10 kPa (larger than 6 kPa), the water can pass through after N₂ treatment.

Based on the above discussion, we have added more detailed discussion about the intrusion pressure of membrane in the revised SI as follow:

Table S2. Variation of the oil and water intrusion pressure of PPFM-0.5 under CO₂/N₂ stimulation

	Water	n-Hexane	Toluene	Isooctane
Dried state ^a	17 kPa	0 kPa	0 kPa	0 kPa
In presence of N ₂ ^b	6 kPa	0 kPa	0 kPa	0 kPa
In presence of CO ₂ ^c	0 kPa	24 kPa	22 kPa	18 kPa

^a The dried state is realized by continuously purging surface with N₂ for a long time. ^b N₂ bubbling time is 20 min. ^cCO₂ bubbling time is 10 min.

As shown in the Table S2, all the oil intrusion pressures of PPFM-0.5 in dried state are 0 kPa while the water intrusion pressure is 17 kPa, which is due to hydrophobicity of membrane surface. After treated with N₂ bubbling for 20 min, the oil intrusion pressures of PPFM-0.5 remain unchanged while the water intrusion pressure decreases from 17 kPa to 6 kPa. This is mainly because CO₂ is not completely removed from water in this case, keeping some hydrophilic and protonated groups still existing in the membrane.

Comment 4:

• The captions in the figures such as Fig S16 don't mention what gap was used to prepare the membranes. All figure and tables giving data about membranes should specify enough data so that the reader can understand what membrane was used. This data should include the gap used.

Response 4:

Thanks for your useful suggestion. We have added the gap used for all figure and tables in the revised manuscript and SI.

Comment 5:

Most of the issues are to do with the English language.

- line 43 “inconspicuous” should be “inadequate”
- many places: “resonsive” should be “responsive”
- line 113: “enlargement” should be “scale-up”
- lines 125 and 150: “inner” should be “inside”
- the paragraph from lines 329 to 376 is far too long. Break it into two paragraphs after ‘reusability’ on line 353.
- SI line 228. What does “wathet” mean? I don’t think that’s a word in the English language.

Response 5:

Thanks for your comment. We apologized for our mistakes. We have corrected them in the revised manuscript and SI. Moreover, the language of the manuscript has been edited by a professional technical editing service. The certificate of professional editing is listed as follows:

Reviewer #3:

General Comments:

Smart membranes have been reported broadly, but most of the membranes are responsive toward stimuli such as pH, salt, and humidity, and temperature, etc. There are relatively less reports in gas-induced responsiveness, which makes this work an interesting article focusing on CO₂-responsive membrane for separating various oil/water systems. Exploration of the CO₂-responsive polymers into membrane separation field is indeed very attractive. Considering the large membrane size and efficiency in tuning surface properties of their membranes, the method developed in this work for fabricating the CO₂-responsive membrane is impressive. Under CO₂/N₂ stimulation, the membrane shows excellent switching wettability in comparison to other reported membranes, resulting in outstanding separation efficiency (>99.9%) for various oil/water systems and self-cleaning performance. The structural and chemical properties of the prepared membranes are thoroughly characterized, and the reviewer appreciates that the results are well backed up by theoretical simulations. Overall, I am in favor of its publication after addressing the following comments:

Response:

Many thanks for the reviewer's positive comments and useful suggestions. According to the reviewers' suggestions, some necessary discussions were added in the revised paper. Meanwhile, some of the wording and language errors were also corrected.

Specific comments:

Comment 1:

In the introduction on CO₂-responsive polymers, the list of references seems incomplete and the important contributions of groups like the P. Theato group (University of Hamburg) and the Q. Yan (Fudan University) are not mentioned. This should be extended.

Response 1:

Thanks for your comment. We have added the works from P. Theato group and the Q. Yan in the revised manuscript as follows:

“28. S.J. Lin, J.J. Shang, P. Theato. Facile fabrication of CO₂-responsive nanofibers from photo-crosslinked poly(pentafluorophenyl acrylate) nanofibers. *ACS Macro Lett.* **7**, 431-436 (2018).

29. S.J. Lin, P. Theato. CO₂-responsive polymers. *Macromol. Rapid Commun.* **34**, 1118-1133 (2013).

30. Q. Yan, Y. Zhao. Block copolymer self-assembly controlled by the “green” gas stimulus of carbon dioxide. *Chem. Commun.* **50**, 11631-11641 (2014).

31. Q. Yan, Y. Zhao. CO₂-stimulated diversiform deformations of polymer assemblies. *J. Am. Chem. Soc.* **135**, 16300-16303 (2013).”

Comment 2:

What's the operating temperature for separation of different oil/water systems? Because the temperature can also impact on the deprotonation process of amine groups

of PDEAEMA.

Response 2:

Thanks for your useful comment. The temperature does have a great influence on the deprotonation process. The operating temperature for all oil/water separation experiments is 25 °C. We have added the operating temperature in the revised manuscript which has been added to the manuscript.

Comment 3:

Authors claim that they exploited capillary force-driven confinement self-assembling (CFCS) strategy to prepare a scalable and stable CO₂-resonsive membrane for smart separation of various oil/water systems. While the mechanism is intriguing, it is not fully clear to me how to define confinement, as this is highly related to the dimension of pore dimensions. Authors would want to elaborate with more details in this regard.

Response 3:

Thanks for your good comment. The confinement we proposed mainly refers to the gap width formed by two pieces of superimposed acrylic plates. The micro-scale gap width can largely impact the transport of the copolymer solution, which finally determines the coating thickness of as-prepared membrane. To facilitate the understanding, we have added the corresponding descriptions about confinement we proposed in the revised manuscript as follows:

“After that, 10 wt% of the copolymer solution (THF as solvent) was slowly injected into the edge of the gap. **Then, under capillary force, the solution was spread in confined space formed by gap width, and self-assembly occurred in situ on the surface and inside of the fabric.** After thermal treatment, a PMMA-co-PDEAEMA-coated fabric membrane, namely, PPFM, was successfully constructed.”

Comment 4:

Authors should add the error bars for all membrane separation measurements, such as that in Fig. 5b, 5e, etc.

Response 4:

Thanks for your comment. We have added the error bars for all membrane separation measurements in the revised manuscript and SI.

Comment 5:

The two-step separation process for multiphase emulsions system is interesting, but author just test two emulsions. Other emulsions used in this work should be also tested. Moreover, why not simply put all data (separation efficiency in Fig. 5B and emulsion content in Fig. S29) in the same panel. This would easily allow readers to grasp what's going on.

Response 5:

Thanks for your useful suggestion. We have added other multiphase emulsions tests in the revised manuscript as follows. We have also merged data of Fig. 5B and in Fig. S29 into the same panel.

Fig. S33 Emulsion content in filtrates of PPFM-0.5 with gap width of 150 μm in 1st and 2nd step operation. The emulsion content is oil content for O/W emulsion and water content for W/O emulsion, respectively.

Comment 6:

As author state, CO₂ is directly bubbled into emulsion solution during test. How about the effect of CO₂ bubbling on stability of emulsion? This is should be investigated.

Response 6:

Thanks for your good comment. In order to investigate the stability of emulsion under CO₂ bubbling, we directly bubbled CO₂ into O/W and W/O emulsions for 3h, and measured their mean diameters every 30 min. As shown in Fig. S25, there are no changes in the mean diameters of all emulsions in the whole process, confirming that CO₂ bubbling has little effect on the stability of emulsions. We have added these figures and related discussions in the revised SI as follows:

Fig. S25 Variation of droplet sizes of various O/W and W/O emulsions under CO₂ stimulation. (A) n-Hexane, (B) Isooctane, (C) Toluene, (D) silicone oil.

The stability of emulsion under CO₂ stimulation is of importance to emulsion separation. As shown in Fig. S25, after bubbling CO₂ into O/W and W/O emulsions for 3h, there are no changes in the mean diameters of all emulsions in the whole process, confirming that CO₂ bubbling has little effect on the stability of emulsions.

Comment 7:

In the Fig. 3 C, D, there is lack of scale bar. Quality of Fig. 31-35 in SI are needed to be improved, the scale bars are difficult to see clearly.

Response 7:

Thanks for your comment. We have added clear scale bars of Fig. 3 C, D and Fig. 31-35 in SI in the revised manuscript and SI.

Comment 8:

Language and expression of this work should be further improved.

Response 8:

Thanks for your comment. The language of the manuscript has been edited by a professional technical editing service. The certificate of professional editing is listed as follows:

This document certifies that the manuscript

Scalable and unprecedentedly wettability-switchable CO₂-responsive membrane for smart separation of various oil/water systems

prepared by the authors

Yangyang Wang, Shaokang Yang, Jingwei Zhang, Zhuo Chen, Bo Zhu, Jian Li, Shijing Liang, Yunxiang Bai, Jianhong Xu, Dewei Rao, Liangliang Dong*, Chunfang Zhang, Xiaowei Yang

was edited for proper English language, grammar, punctuation, spelling, and overall style by one or more of the highly qualified native English speaking editors at SNAS.

This certificate was issued on **January 13, 2023** and may be verified on the SNAS website using the verification code **2BD7-5036-8534-C9F4-E33B**.

Neither the research content nor the authors' intentions were altered in any way during the editing process. Documents receiving this certification should be English-ready for publication; however, the author has the ability to accept or reject our suggestions and changes. To verify the final SNAS edited version, please visit our verification page at secure.authorservices.springernature.com/certificate/verify.

If you have any questions or concerns about this edited document, please contact SNAS at support@as.springernature.com.

Reviewers' Comments:

Reviewer #1:

Remarks to the Author:

The manuscript can be accepted.

Reviewer #2:

Remarks to the Author:

The responses to my comments and the comments of the other referees are quite satisfactory. I recommend that the paper be accepted for publication.

Reviewer #3:

Remarks to the Author:

Authors have nicely addressed the reviewer's concerns and the revised work is recommended for publication as it is.

Response to comments

Reviewer #1

The manuscript can be accepted.

Response:

Thanks for your time to review our manuscript. We express our sincere thanks for your valuable comments to improve the quality of our work and also for your kind approval to accept our manuscript for publication.

Reviewer #2:

The responses to my comments and the comments of the other referees are quite satisfactory. I recommend that the paper be accepted for publication.

Response:

Thanks for your time to review our manuscript. We appreciate your high recognition and express our sincere thanks for your valuable comments to improve the quality of our work, and also for your kind approval to accept our manuscript for publication.

Reviewer #3:

Authors have nicely addressed the reviewer's concerns and the revised work is recommended for publication as it is.

Response:

Thanks for your time to review our manuscript. We express our sincere thanks for your valuable comments to improve the quality of our work and also for your kind approval to accept our manuscript for publication.